# Status and future prospects for mobile phone-enabled diagnostics in Tanzania

**Ndyetabura O. Theonest**[1]\*, **Kennedy Ngowi**[1], **Elizabeth R. Kussaga**[1], **Allen Lyimo**[1], **Davis Kuchaka**[1], **Irene Kiwelu**[1,2], **Dina Machuve**[3], **John-Mary Vianney**[3], **Julien Reboud**[4], **Blandina T. Mmbaga**[1,2], **Jonathan M. Cooper**[4], **Joram Buza**[3]

1 Kilimanjaro Clinical Research Institute, Moshi, Tanzania, 2 Kilimanjaro Christian Medical University College, Moshi, Tanzania, 3 Schools of Life Sciences and Bioengineering, Nelson Mandela African Institution of Science and Technology, Arusha, Tanzania, 4 Division of Biomedical Engineering, Mazumdar-Shaw Advanced Research Centre, University of Glasgow, Glasgow, United Kingdom

\* t.ndyetabura@kcri.ac.tz

**Data Availability Statement:** The de-identified data pertaining to this survey is publicly available through the following https://www.ebi.ac.uk/biostudies/studies/S-BSST1212.

## Abstract

### Introduction

Diagnosis is a key step towards the provision of medical intervention and saving lives. However, in low- and middle-income countries, diagnostic services are mainly centralized in large cities and are costly. Point of care (POC) diagnostic technologies have been developed to fill the diagnostic gap for remote areas. The linkage of POC testing onto smartphones has leveraged the ever-expanding coverage of mobile phones to enhance health services in low- and middle-income countries. Tanzania, like most other middle-income countries, is poised to adopt and deploy the use of mobile phone-enabled diagnostic devices. However, there is limited information on the situation on the ground with regard to readiness and capabilities of the veterinary and medical professionals to make use of this technology.

### Methods

In this study we survey awareness, digital literacy and prevalent health condition to focus on in Tanzania to guide development and future implementation of mobile phoned-enable diagnostic tools by veterinary and medical professionals. Data was collected using semi-structured questionnaire with closed and open-ended questions, guided in-depth interviews and focus group discussion administered to the participants after informed consent was obtained.

### Results

A total of 305 participants from six regions of Tanzania were recruited in the study. The distribution of participants across the six regions was as follows: Kilimanjaro (37), Arusha (31), Tabora (68), Dodoma (61), Mwanza (58), and Iringa (50). Our analysis reveals that only 48.2% (126/255) of participants demonstrated significant awareness of mobile phone-enabled diagnostics. This awareness varies significantly across age groups, professions and geographical locations. Interestingly, while 97.4% of participants own and can operate

**Funding:** This work was supported by funds from the Engineering and Physical Sciences Research Council (EPSRC), UKRI grant reference: EP/T029765/1, JC. This work is also part of a project that has received funding from the European Union's Horizon 2020 research and innovation programme under grant agreement No. 101057251 (DIDIDA), as well as from United Kingdom Research and Innovation Innovate UK (10052860). The funder had no role in study design, data collection and analysis, decision to publish, or preparation of the manuscript.

**Competing interests:** The authors have declared that no competing interests exist.

a smartphone, 62% have never utilized their smartphones for health services, including disease diagnosis. Regarding prevalent health condition to focus on when developing mobile phone -enabled diagnostics tools for Tanzania; there was disparity between medical and veterinary professionals. For medical professionals the top 4 priority diseases were Malaria, Urinary Tract Infections, HIV and Diabetes, while for veterinary professionals they were Brucellosis, Anthrax, Newcastle disease and Rabies.

## Discussion

Despite the widespread ownership of smartphones among healthcare providers (both human and animal), only a small proportion have utilized these devices for healthcare practices, with none reported for diagnostic purposes. This limited utilization may be attributed to factors such as a lack of awareness, absence of policy guidelines, limited promotion, challenges related to mobile data connectivity, and adherence to cultural practices.

## Conclusion

The majority of medical and veterinary professionals in Tanzania possess the necessary digital literacy to utilize mobile phone-enabled diagnostics and demonstrate readiness to adopt digital technologies and innovations to enhance diagnosis. However, effective implementation will require targeted training and interventions to empower them to effectively apply such innovations for disease diagnosis and other healthcare applications.

### Author summary

Mobile phone-enabled diagnostics have emerged as a promising solution to overcome barriers to healthcare access in low-resource settings such as Tanzania. Leveraging the widespread penetration of mobile phones there is a potential to empower healthcare providers, improve diagnostic accuracy, and enhance patient outcomes. However, their effective integration into clinical practice necessitates an understanding of the prevailing context, including technological infrastructure, regulatory frameworks, and socio-economic factors. Our study reveals a notable level of digital literacy among medical and veterinary professionals in Tanzania, indicating a favourable environment for the adoption of mobile phone-enabled diagnostics. Despite this, awareness gaps persist, hindering widespread utilization. Furthermore, logistical and infrastructural challenges pose barriers to implementation, particularly in rural and underserved areas. Ethical considerations, including patient privacy and data security, also merit attention to ensure the responsible deployment of these technologies. In order to capitalize on the potential of mobile phone-enabled diagnostics, several key areas warrant further exploration. Targeted awareness campaigns and specialized training initiatives are needed to enhance the uptake and utilization of these tools among healthcare providers. Research focusing on rural populations is imperative to address disparities in healthcare access and delivery. Ethical frameworks and regulatory guidelines must be developed to govern the use of mobile health technologies, balancing innovation with privacy and security concerns. Collaborative efforts between policymakers, healthcare professionals, technology developers, and community stakeholders are essential to drive sustainable implementation and scalability. By addressing these critical gaps, mobile phone-enabled diagnostic tools can be implemented in

Tanzania, thereby improving healthcare access, diagnostic accuracy, and ultimately, patient outcomes in Tanzania.

## Introduction

More than two-thirds of human and animal healthcare professionals own and use their smartphones for phone calls regularly [1, 2]. Whilst the application of smartphones is increasing across the world including low- and middle -income countries (LMICs), there is lack of information in Tanzania with regard to the use of smartphones for disease diagnosis among human and animal healthcare professionals. The use of smartphones for disease diagnosis purposes is particularly important in LMICs where most of diagnostic capabilities including professionals, equipment, and reliable utilities are concentrated in big cities and, therefore, not accessible to a majority of the population who live in remote areas [3]. Point-of-care (POC) diagnostics have been developed to extend clinical testing capabilities to the point of need particularly in community settings and in remote areas. Based on the World Health Organization, an ideal POC diagnostic must be Affordable, Sensitive, Specific, User-friendly, Rapid/Robust, Equipment-free and Deliverable (ASSURED) [4]. While most available POC and those being developed can meet many of the above requirements, the "Equipment-free" requirement has been hard to meet because attaining very low detection limits (nanograms) needed for detection of most biomarkers requires sensitive, accurate and reliable instrumentation, which makes them less cost effective and less mobile. The technological advancement of the digital age has led to a revision of the ASSURED criteria to also include Real-time connectivity, ease of specimen collection, (REASSURED) [5].

As capabilities for smartphones increase in view of features such as 3G to 5G, WiFi, Bluetooth, and near field communication (NFC) [6], their adaptability for diagnosis also increases. Some of the functionalities of mobile phones which are being used for diagnosis include power, as a source for heating in polymerase chain amplification (PCR) or other, more suitable isothermal amplification [7], data transmission [8], use of cameras to capture screenshots from ultrasound [9], or images of paper-based assay and send them for analysis, often combined in a single platform [6]. The smartphone camera has also enabled colorimetric methods and sensitive techniques such as fluorescence and electrogenerated chemiluminescence [10–12]. Examples of currently running smartphone-enabled diagnostic tests include the "Dongle", a POC enzyme linked immunosorbent assay (ELISA) for HIV and syphilis [13], smartphone digital microscopy for diagnosis of helminths infections [14], nucleic acid detection systems for malaria [15], where the smartphone acts as power source, a computer for controlling the diagnostic test, results readout, decision support, and data storage (locally or to the cloud).

Tanzania, like many other LMICs, appears ready to implement smartphone-enabled diagnostics in day to day medical and veterinary healthcare services [16]. However, baseline information is still missing with respect to awareness of mobile phone use for health, ability to operate smartphone functionalities that are relevant for diagnosis and prevalent health condition to be targeted when developing mobile phone-enabled diagnostics devices. Implementation of mobile phone-enabled diagnostics represents a transformative step forward in healthcare delivery and disease management in Tanzania including enhancing disease surveillance efforts by enabling rapid detection and response to outbreaks; ultimately contributing to better public health outcomes, promoting personalized medicine; personalized approach can lead to more effective treatment strategies and improved patient outcomes. Furthermore, mobile phone-enabled diagnostics are expected to support healthcare system strengthening

through improved efficiency, reduced diagnostic turnaround times, enhanced patient engagement, and better resource management, mobile phone-enabled diagnostics will pave the way for advancements in telemedicine services in Tanzania, allowing for remote consultations, diagnosis verification by specialists located elsewhere, and continuous monitoring of patients with chronic conditions. The data generated through mobile phone-enabled diagnostics present valuable opportunities for research in epidemiology, public health interventions, treatment efficacy studies, and health behaviour analysis within the Tanzanian context.

To gather baseline information, we conducted a survey study aimed at assessing the current utilization of mobile phone-enabled diagnostics among human and animal healthcare professionals in Tanzania, identifying the challenges and barriers faced in implementing such diagnostics in the healthcare system, and exploring potential future prospects for integrating them into routine healthcare practices. Although our study was more exploratory in nature, we anticipated varying levels of awareness and adoption among healthcare professionals in different regions of Tanzania. We also expected barriers such as infrastructure limitations and technological literacy to hinder widespread implementation, but we see the potential for targeted interventions and capacity building efforts to overcome these barriers. In this study, we sought to obtain this information using a structured questionnaire, in-depth interviews and focus group discussion targeting medical and veterinary personnel from different regions of Tanzania. Such information could enable researchers and policy makers to prioritize activities to build capacity and facilitate effective deployment of mobile phone enabled disease diagnostic tools in future.

## Methods

### Study design

The cross-sectional survey study involved the collection of information on the awareness, literacy and prevalent health condition for mobile phone-enabled disease diagnosis in Tanzania, among medical and animal healthcare providers. To collect this information, we used a structured questionnaire, in-depth interviews and focus group discussions.

### Study area

This study was conducted in six regions of Tanzania mainland. Tanzania is divided in six administrative zones and one region was selected from each zone except for the Northern zone, which included 2 regions (Kilimanjaro and Arusha regions), due to fact that many mHealth research studies and publications in Tanzania have been done from this zone [17]. Other zones with selected regions in parentheses were: Southern (Iringa region), Western (Tabora region), Lake (Mwanza region), Central (Dodoma region), and Coastal (Dar es Salaam region) (**Fig 1**). The selected regions, hospitals and health facilities had professionals (based on literacy level) capable of providing constructive opinion on the use of mobile phone for health purposes. The selected regions also had veterinary clinics and laboratories, where data was collected.

### Sample size determination

The determination of the sample size was conducted through a comprehensive assessment of the estimated total of healthcare workers (66,348) and animal healthcare workers (8,943) in Tanzania, resulting in a combined population of 75,291 individuals [19]. This calculation took into account the proportion of smartphone users within the study population, which was estimated at approximately 75%, supported by existing literature and data on smartphone

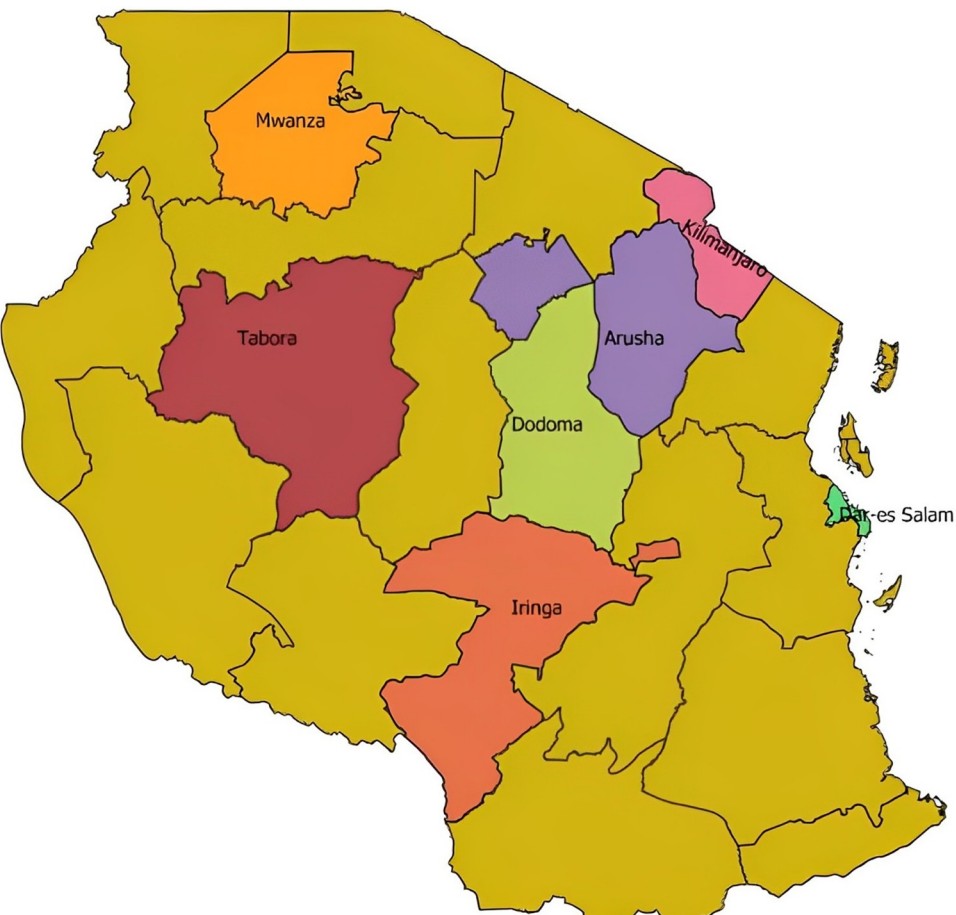

**Fig 1. Map of Tanzania indicating surveyed regions marked in different colours with their names added in the map.** The survey was conducted to assess awareness, literacy and prevalent health condition for mobile phone-enabled diseases/condition diagnosis among medical and veterinary professionals. The map was created using QGIS [18], an official project of the Open-Source Geospatial Foundation (OSGeo) licensed under the Creative Commons Attribution 4.0. The shapefile is available from **https://energydata.info/dataset/tanzania-region-district-boundary-2012**.

penetration in Tanzania [20]. To ensure the external validity and generalizability of the findings, the sample size was further adjusted for each sampling region, resulting in a total of 305 instead of 288 individuals. The allocation of the overall sample size across regions was not arbitrary. Instead, it was meticulously designed to ensure adequate representation of the diverse geographical areas within Tanzania. Each region received a proportionate share of the sample size, allowing for the capture of variations in healthcare infrastructure, access to technology, and other contextual factors that could influence the adoption and implementation of mobile phone-enabled diagnostics.

## Study subjects

The study participants were recruited from healthcare workers (including Doctors, Nurses, Radiologists, Laboratory technicians, Biomedical engineers) and animal healthcare workers (including Veterinary doctors, Veterinary technicians), based on their professional skills and their role at the health facilities or veterinary clinics.

### Ethical statement

Ethical approval for the study was granted by the Kilimanjaro Christian Medical University College (KCMUCo) Ethics Committee (Certificate # 2504); National Institute for Medical Research (NIMR), Tanzania (NIMR/HQ/R.8a/Vol.IX/3614) and Presidents Office Regional Administration and Local Government (Kumb.Na.AB.307/323/01). Written consent for study participation was obtained from each study participant using forms translated into Swahili. Participation in the study was voluntary; for confidentiality, each participant was assigned a unique identification number.

### Data collection

An assessment of awareness, literacy, opinion about mobile phone-enabled diagnostics among medical and veterinary workers was done using guided semi-structured questionnaires, in-depth interview and focused group discussion administered to the study participant by NT, KN, EK, AL, DK, IK, DM, and JV (letters stands for initials of first letters for first name and surname of authors). The tools were also used to determine demographic factors influencing use and adoption. Open-ended questions in a focused group discussion allowed to collect the feedback and opinion from the participants about their expectations and experience in their own narrative. Data were collected and entered by the individual interviewer into the open data kit (ODK) software (version v1.28.4) [21] installed in tablets at the point of interview. At the end of the interview for every study participant, the collected data and information were reviewed by a different investigator to assess the quality and integrity of the collected data and information prior to submission to the local server maintained at Kilimanjaro Clinical Research Institute for aggregation.

### Statistical analysis

Our analysis used a sequential explanatory design whereby the quantitative data were descriptively analysed using python 3.6.0 [22] to assess awareness and literacy of mobile phone-enabled diagnostics among human and animal healthcare professionals.

To understand the proposed prevalent health condition to be considered for diagnosis using smartphone enabled devices in the future, the narrative information from the open-ended responses to the questionnaire were summarised thematically to link the analysis to the quantitative data. In order to allow for inductive discovery and subsequent interpretation of themes and sub-themes, all of the collected responses were subsequently analyzed using thematic analysis approach [23]. Our analysis first involved reading through all the responses to identify and sort them into key themes and sub-themes. After several iterations among investigators, DM, KN and TN separately developed codes for themes and sub-themes for 20% consensus transcripts responses. In conducting the thematic analysis, we adhered to rigorous methodological standards to ensure the credibility and trustworthiness of our findings. Data analysis was approached systematically, involving a thorough familiarization with the data, followed by the generation of initial codes. Data sources was triangulation to corroborate emerging themes, ensuring robustness in our analysis. To enhance the credibility of our findings, member checking was employed, allowing participants the opportunity to verify the accuracy and interpretation of their responses. In-depth interview field notes, memoranda, and transcripts were organized and entered in NVivo software version 12 [24] hosted at Kilimanjaro Clinical Research Institute to facilitate further analysis, including comparison of the relative theme and sub-theme frequencies.

To establish the correlation between different variables and our primary variable of interest (awareness of mobile phone-enabled diagnostics), we performed binary logistic or

multinomial logistic regression analysis to calculate crude and adjusted odds ratios using SPSS software (Version 26). We focused on human healthcare workers only due to their relatively large number (n = 257; 84.3%).

## Results

### Sample characteristics and demographics

A total of 305 healthcare workers were interviewed: 257 (84.26%) human healthcare and 48 (15.74%) animal healthcare. The majority of human health facilities and veterinary clinics visited for interview were from urban areas (80.7%), referral regional hospitals (36.4%), general workers (staff) (57.4%), nurses (33.1%), diploma holders (37.7%), male (51.5%), and those who were aged between 25 and 34 years (49.2%), (**Table 1**).

### Thematic analysis

Thematic analysis of the qualitative data revealed several key themes regarding the status and future prospects of mobile phone-enabled diagnostics in Tanzania. Among the prominent findings were barriers to adoption, including limited access to smartphones rural areas and mobile data bundles. Additionally, the analysis identified opportunities for intervention, such as targeted educational campaigns to enhance digital literacy, infrastructure improvements to expand access to reliable internet connectivity, and collaborations with local healthcare providers to integrate mobile diagnostics into existing healthcare systems. Overall, the thematic analysis provided valuable insights into the challenges and opportunities associated with mobile phone-enabled diagnostics in Tanzania, informing potential strategies to address existing barriers and maximize the benefits of mobile phone-enabled diagnostics in improving healthcare access.

### Ownership and use of smartphone

Almost all study participants owned one or more smartphones (97.4%). The model or type of smartphone most owned was Techno (33.1%), and the majority 74.4% have owned a smartphone for more than five years. With regard to the use of smartphone for healthcare practices, 68.2% had never utilized their smartphone for healthcare practice, while 10.2% of them have used their smartphone to access medical information for less than one year. The reasons for use of smartphone in healthcare practice were: as a source of information on medicines, to access medical websites, to review guidelines and protocols related to care, for procedure documentation, and as a source of patient education materials, while none reported use for diagnostic purposes. 68.0% reported using their smartphones for more than five hours a day, while only 3.6% reported using their smartphone for less than one hour per day (**Table 2**).

During in-depth interview, some of the study participants said:—*"I have owned different brands of smartphone for many years, I often use my phone for making calls and social media. I have never used my phone for diagnostic purposes. Use of phone for diagnostics will be a good thing because I will be able know the cause of diseases of my patient and hence give appropriate care". A senior nurse at Kilimanjaro Christian Medical Centre.*

*A Laboratory technician, in Mwanza said,*

*"I don't know how to use my phone for diagnostic purposes, I think before deployment of the technology it will be better to adequately invest in educating both healthcare professionals and the general population about the benefits and limitations of mobile diagnostics, this will be crucial for successful implementation"*

**Table 1. Demographic characteristics of the study population.**

| Characteristics | No (n = 305) | % |
|---|---|---|
| **Sex** | | |
| Male | 157 | 51.5 |
| Female | 148 | 48.5 |
| **Age group (years)** | | |
| 18–24 | 12 | 3.9 |
| 25–34 | 150 | 49.2 |
| 35–44 | 69 | 22.6 |
| 45–54 | 50 | 16.4 |
| 55–64 | 22 | 7.2 |
| Over 65 | 2 | 0.7 |
| **Education** | | |
| Ordinary Level | 12 | 3.9 |
| Advanced Level | 11 | 3.6 |
| Diploma | 115 | 37.7 |
| Advanced Diploma | 20 | 6.6 |
| Bachelor Degree | 114 | 37.4 |
| Masters and PhD | 33 | 10.8 |
| **Occupation** | | |
| Nurse | 101 | 33.1 |
| Medical Doctor | 82 | 26.9 |
| Veterinary Doctor | 11 | 3.6 |
| Veterinary Technician | 20 | 6.6 |
| Information and Communication Technology Personnel | 16 | 5.2 |
| Radiologist | 9 | 2.9 |
| Biomedical Engineer | 6 | 2 |
| Others | 60 | 19.7 |
| **Job Title** | | |
| Staff | 175 | 57.4 |
| In charge | 40 | 13.1 |
| Medical Supervisor | 36 | 11.8 |
| Supervisor | 23 | 7.5 |
| Others | 31 | 10.2 |
| **Level of Facility** | | |
| Regional Referral Hospital | 111 | 36.4 |
| District | 101 | 33.1 |
| Zonal Referral Hospital | 54 | 17.7 |
| National | 15 | 5 |
| Village | 8 | 2.6 |
| Other | 16 | 5.2 |
| **Regions** | | |
| Arusha | 31 | 10.2 |
| Kilimanjaro | 37 | 12.1 |
| Tabora | 68 | 22.3 |
| Dodoma | 61 | 20 |
| Mwanza | 58 | 19 |
| Iringa | 50 | 16.4 |
| **Location of Facilities** | | |

*(Continued)*

**Table 1.** (Continued)

| Characteristics | No (n = 305) | % |
|---|---|---|
| Urban | 246 | 80.7 |
| Rural | 59 | 19.3 |

## Proposed prevalent health condition for diagnosis using smartphone in future

Participants from human and animal healthcare workers expressed considerable differences on their priorities, mainly linked to their areas of professional practice. The majority of human healthcare workers prevalent health condition for mobile phone-enabled diagnostic tools were malaria (94%), urinary tract infections (85%), HIV (85%), COVID-19 (84%) and diabetes (75%) (**Fig 2**). On the other hand, animal healthcare workers priority diseases were Brucellosis (87.5%), Anthrax (70%), Newcastle diseases (60.4%), Rabies (58%), and Bovine pneumonia (56.3%) (**Fig 3**). Regarding prevalent health condition, some of respondent said: -

*A district veterinary officer in Mwanza said*: -

*"In the animal sector, we are almost not carrying out diagnostics to identify what makes animals get sick, if our mobiles can be used for diagnostic of animal diseases, definitely we are going to welcome the innovation, specifically I think Mobile diagnostics could serve as a valuable tool for animal diseases surveillance such as anthrax, brucellosis and respiratory diseases but also in outbreak response, enhancing Tanzania's capacity to address public health emergencies and zoonotic diseases."*

*A physician at Kiteto region hospital, Tabora said*: -

*"Mobile phone diagnostics will empower us to provide faster and more accurate healthcare services, especially for malaria and urinary tract infection diagnosis, diseases which are major problem in Tabora region but their differential diagnostic is a major problem due to unspecific clinical presentation."*

## Discussion

Increasing use of smartphone applications in healthcare practices among healthcare professionals has gained wide acceptance in many countries globally. The percentage of healthcare professionals using smartphones is exponentially increasing both globally and in Tanzania [25, 26]. In our study, almost all health providers owned one or more smartphones **Table 2** but only 30.5% had utilized their smartphone for healthcare practices, and none have used their smartphones for diagnostics purposes. The low utilization of smartphones for healthcare practices in Tanzania underscores a complex interplay of socio-economic, infrastructural, regulatory and cultural factors that warrant further examination. One potential explanation for this phenomenon is the significant disparities in access to smartphones and reliable internet connectivity across different demographic groups within the country. Moreover, concerns regarding data privacy and security, compounded by a lack of awareness and trust in mobile health technologies, may contribute to hesitancy among both healthcare providers and patients in embracing smartphone-based diagnostics. Additionally, cultural attitudes towards healthcare delivery, including preferences for in-person consultations and traditional healing practices, may also influence the adoption of digital health solutions. To address these challenges and

**Table 2. Ownership and utilization of smartphones by medical and veterinary healthcare workers.**

|  | No (n = 305) | % |
|---|---|---|
| **Smartphone model** | | |
| Nokia | 8 | 2.6 |
| iPhone | 33 | 10.8 |
| Techno | 101 | 33.1 |
| Samsung | 80 | 26.3 |
| Infinix | 41 | 13.4 |
| Others | 42 | 13.8 |
| **Duration of owning Smartphone** | | |
| Never | 6 | 2 |
| 1 year | 10 | 3.3 |
| 2 years | 8 | 2.6 |
| 3 years | 19 | 6.2 |
| 4 years | 11 | 3.6 |
| 5 years | 24 | 7.9 |
| More than 5 years | 227 | 74.4 |
| **Awareness of mHealth** | | |
| Yes | 164 | 53.8 |
| No | 132 | 43.2 |
| Maybe | 9 | 3 |
| **Apps used for video calling** | | |
| WhatsApp only | 203 | 66.6 |
| WhatsApp & Messenger | 26 | 8.5 |
| Zoom only | 22 | 7.2 |
| WhatsApp & zoom | 12 | 3.9 |
| WhatsApp, Imo & team | 9 | 3 |
| Others | 29 | 9.5 |
| **Apps for documentation** | | |
| Google Keep | 131 | 43 |
| Microsoft Word | 66 | 21.6 |
| Microsoft Word & Google Keep | 28 | 9.2 |
| Microsoft Word, Google Keep & Others | 8 | 2.6 |
| Others | 55 | 18 |
| Microsoft Word & Others | 10 | 3.3 |
| Google Keep & Others | 7 | 2.3 |
| **Duration of using smartphone per day** | | |
| Less than 1 hour | 11 | 3.6 |
| 1 hour | 16 | 5.2 |
| 3 hours | 31 | 10.2 |
| 4 hours | 21 | 6.9 |
| 5 hours | 18 | 5.9 |
| Over 5 hours | 208 | 68.2 |
| **Duration of usage of mHealth applications** | | |
| Never | 212 | 69.5 |
| 1 Year | 32 | 10.5 |
| 2 Years | 13 | 4.3 |
| 3 Years | 14 | 4.6 |
| 4 Years | 8 | 2.6 |

(*Continued*)

**Table 2.** (Continued)

|  | No (n = 305) |  | % |
|---|---|---|---|
| 5 Years | 7 |  | 2.3 |
| More than 5 Years | 19 |  | 6.2 |

unlock the full potential of mobile phone-enabled diagnostics in Tanzania, multifaceted strategies are needed. These may include targeted educational campaigns to enhance digital literacy and promote trust in mobile health technologies, infrastructure investments to expand access to affordable smartphones and internet connectivity, collaborations with local healthcare providers to integrate mobile diagnostics into existing service delivery models, and policy interventions to establish clear regulatory frameworks ensuring patient privacy and data security [27].

The most frequent use of smartphone was for making regular calls and social activities such as streaming and watching movies (98.7%), the most commonly used applications (app) were WhatsApp and YouTube. This is evidenced by response from a quality officer at one of referral hospitals who participated in the in-depth interview, he said: -

> "I have never used my phone for diagnostics, I only use my phone for making calls and on social media especial WhatsApp and YouTube, I don't know which app and how to use for diseases diagnostics. But I think the use of mobile apps for diagnostics will reduce diagnostic burden of sample at our clinical laboratory and improved the overall quality of healthcare services we provide."

Understanding which smartphone applications are commonly used by healthcare professionals is crucial for the development of digital and mobile phone-enabled tools for disease diagnosis. The finding that WhatsApp and YouTube are the most commonly used applications, comprising 66.6% usage, underscores the significance of tailoring innovations to

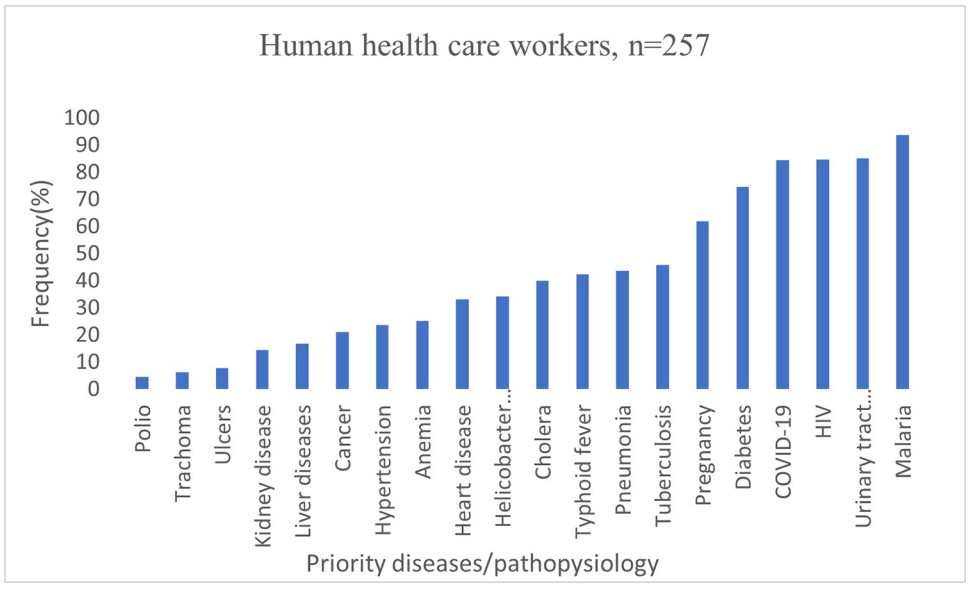

**Fig 2. Prevalent health condition for mobile phone enabled diagnostics amongst human healthcare workers in Tanzania.**

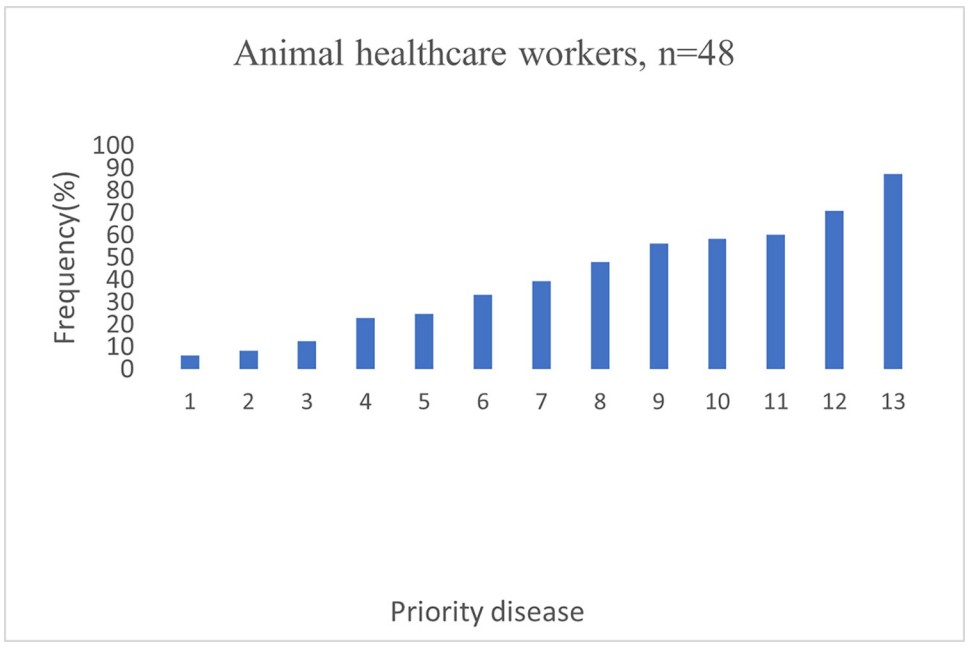

**Fig 3. Prevalent health condition for mobile phone enabled diagnostics amongst animal healthcare workers in Tanzania.**

integrate disease diagnostic tools with these widely utilized platforms. By aligning new technologies with existing user habits, researchers can greatly enhance the adoption and usage of these tools within healthcare settings. This integration not only streamlines access to diagnostic resources but also promotes seamless integration into healthcare professionals' workflows, ultimately improving patient care and outcomes. Interrogated respondents about what smartphone application they use frequently, a senior nurse said: -

> *"For the past five years, I have used different smartphone brands including sumsang, Vivo, iPhone and now I am using techno. The apps which I frequently use are WhatsApp, YouTube, TikTok, Instagram and video conference apps I use Zoom and teams only".*

Approximately half of the study participants were aware of mHealth, similar to other studies in LMICs, indicating that similar opportunities and challenges around mhealth could be resonating in such settings [27–29]. Amongst those who used smartphones for accessing medical information, a majority were medical doctors, followed by nurses, hinting on the importance of targeting specific groups when designing and deployment strategy for mobile phone-enabled diseases diagnostic tools. In an in-depth interview with a resident pediatrician at Mawenzi, regional hospital about his awareness on mHealth, he said; -

> *"I am aware of mHealth; infarct I recently finished a research project as a co-investigator aiming at improving adherence to ARV prescription among HIV-pregnant women in Kilimanjaro, the project aimed at using mobile phone to remind patient to take ARVs and notify the doctor that the patient has opened the pill box and hence assuming has taken the pill"*

On the other hand, the number of doctors and nurses is often larger than that of other healthcare personnel and that doctors and nurses interact with patients more frequently than

**Table 3. The correlation of mHealth amongst human healthcare workers in Tanzania (n = 257).**

| VARIABLE | n (%) | mHealth | | Crude OR | | Adjusted OR (P-value <0.1) | |
|---|---|---|---|---|---|---|---|
| | | Yes (%) | No (%) | OR (95% CI) | P-Value | OR (95% CI) | P-Value |
| **Occupation** | | | | | | | |
| Doctors | 80(32.1) | 54(38.8) | 26(23.6) | 0.6(0.3–1.2) | 0.16 | N/A | N/A |
| Nurse | 98(39.4) | 45(32.4) | 53(48.2) | 1.5(0.8–2.8) | 0.18 | N/A | N/A |
| Other | 71(28.5) | 40(28.8) | 31(28.2) | N/A | N/A | N/A | N/A |
| **Sex/Gender** | | | | | | N/A | N/A |
| Male | 117(47) | 66(47.5) | 51(46.4) | 1.0(0.6–1.6) | 0.86 | N/A | N/A |
| Female | 132(53) | 73(52.5) | 59(53.6) | N/A | N/A | N/A | N/A |
| **Level of Education** | | | | | | | |
| Masters/PhD | 24(9.6) | 17(12.2) | 7(6.4) | 0.3(0.1–1.1) | 0.06 | 0.4(0.1–1.7) | 0.22 |
| Degree | 90(36.1) | 53(38.1) | 37(33.6) | 0.5(0.2–1.4) | 0.19 | 1.0(0.3–3.1) | 0.97 |
| A/Diploma | 114(45.8) | 60(43.2) | 54(49.1) | 0.7(0.3–1.7) | 0.41 | 1.2(0.4–3.4) | 0.76 |
| A/O-level | 21(8.4) | 9(6.5) | 12(10.9) | N/A | N/A | N/A | N/A |
| **Location** | | | | | | | |
| Urban | 198(79.5) | 106(76.3) | 33(23.7) | 1.6(0.8–3.0) | 0.15 | N/A | N/A |
| Rural | 51(20.5) | 92(83.6) | 18(16.4) | N/A | N/A | N/A | N/A |
| **Region** | | | | | | | |
| Arusha | 24(9.6) | 22(15.8) | 2(1.8) | 0.8(0.1–4.6) | 0.78 | 0.9(0.1–5.4) | 0.9 |
| Dodoma | 49(19.7) | 18(12.9) | 31(28.2) | 14.6(4.5–48.0) | 0.00 | 15.8(4.5–55.1) | 0.00 |
| Kilimanjaro | 30(12.0) | 18(12.9) | 12(10.9) | 5.7(1.6–20.1) | 0.007 | 8.2(2.1–32.8) | 0.003 |
| Mwanza | 48(19.3) | 13(9.4) | 35(31.8) | 22.9(6.8–77.2) | 0.00 | 26.4(7.2–96.4) | 0.00 |
| Tabora | 60(24.1) | 34(24.5) | 26(23.6) | 6.5(2.0–20.6) | 0.001 | 7.9(2.3–27.4) | 0.001 |
| Iringa | 38(15.3) | 34(24.5) | 4(3.6) | N/A | N/A | N/A | N/A |
| **Age** | | | | | | | |
| <24 | 11(4.4) | 5(3.6) | 6(5.5) | 1.7(0.5–6.2) | 0.42 | N/A | N/A |
| 24–34 | 123(49.4) | 70(50.4) | 53(48.2) | 1.1(0.6–2.0) | 0.83 | N/A | N/A |
| 34–44 | 57(22.9) | 30(21.6) | 27(24.5) | 1.3(0.6–2.7) | 0.52 | N/A | N/A |
| 44–64 | 58(23.3) | 34(24.5) | 24(21.8) | N/A | N/A | N/A | N/A |
| **Role at health facility** | | | | | | | |
| Supervisor/in charge | 45(18.1) | 36(25.9) | 9(8.2) | 0.3(0.1–0.9) | 0.04 | 0.8(0.2–2.9) | 0.72 |
| Medical super user | 31(12.4) | 21(15.1) | 10(9.1) | 0.6(0.2–1.8) | 0.37 | 0.4(0.1–1.3) | 0.12 |
| Staff | 148(59.4) | 68(48.9) | 80(72.7) | 1.5(0.6–3.5) | 0.35 | 1.0(0.4–3.0) | 0.95 |
| Other/director | 25(10.0) | 14(10.1) | 11(10.0) | N/A | N/A | N/A | N/A |
| **Ownership/Access** | | | | | | | |
| Yes | 242(97.2) | 135(97.1) | 4(2.9) | 1.1(0.2–4.8) | 0.94 | N/A | N/A |
| No | 7(2.8) | 107(97.3) | 3(2.7) | N/A | N/A | N/A | N/A |

other professionals, therefore, likely to be a group of healthcare personnel that are looking for a quick means of understanding what is causing diseases to their patient more often Table 3. Additionally, medical doctors were much more aware than other health professionals about the possibilities of using their smartphones for diseases diagnostics and for accessing health information using their smartphone Table 3. This is in contrast with a previous study done in Ghana which found that nurses were the most frequent users of smartphones [30].

Most smartphone users reported owning Techno and Samsung models, although the smartphone vendors market in Tanzania is very dynamic, this has a consequential impact on the type of smartphone frequently owned and their functional capabilities [31]. This could potentially impact the performance and compatibility with medical apps developed. Older

smartphones are likely not able to run 4G/5G specific features, which enhance internet connectivity and data transmission capacity [32].

Although the majority of participants reported having owned a smartphone for more than five years, (and only few never owned one) about half were aware of mHealth and smartphone enabled diagnostics **Table 2**, indicating the need for promoting the use of mHealth among healthcare providers. Interestingly, the majority of respondents reported using their smartphone for more than five hours per day and only a small proportion used them for less than one hour, implying that smartphones could be used at POC, close to a patient/animal. The fact that majority of healthcare professionals own smartphones is an added advantage towards promoting their use for diagnostics and this finding concords with global trends [33, 34]. For example, a survey study conducted by the Accreditation Council for Graduate Medical Education (ACGME) in the USA examined smartphone applications and showed that more than 85% of respondents used a smartphone, of which the iPhone was the most popular (56%), indicating that use of smartphones is increasing globally and model varies considerably in different countries [35].

All respondents were very enthusiastic of the potential of smartphones enabled disease diagnostics in Tanzania. In response to question on whether healthcare professional will be ready to use their smartphone for making diagnostics, a head of department at Kiteto regional hospital in Tabora region said: -

"*Mobile phone-enabled diagnostics have the potential to revolutionize healthcare delivery in Tanzania by bringing essential medical services closer to our clients. However, ensuring the accuracy and reliability of these diagnostic tools need to be critically considered before deployment*".

In a focus group discussion, respondents expressed different opinions on prevalent health conditions, which were related to the area of expertise of the respondent: respondents from the veterinary field mentioned zoonotic disease such *Bacillus anthrax*, *Brucella* species, Rabies, East coast fever, tick borne diseases, as well as Trypanosomiasis as their priority diseases for mobile phone-enabled diseases diagnostics **Fig 3**, while human healthcare professionals had diverse opinions and understanding. Medical doctors appeared to support the use of smartphone-based devices for diagnostics more than nurses and professionals from other departments, as observed in other countries [36].

Prevalent health conditions for mobile phone-enabled diagnostic among human healthcare did not vary regionally and included Malaria, Tuberculosis, Human Immunodeficiency Virus (HIV), Urinary Tract Infections, Pregnancy, Blood pressure, Diabetes, and COVID-19 **Fig 2**. Tanzania is promoting One Health approaches to disease diagnosis and management and has developed and revised its five-years national One Health strategic plan [37], but, respondents in both groups (human and animal healthcare professionals) did not seem to be aware of the linkages between human and animal health and potential for zoonotic diseases **Figs 2 & 3**.

Some respondents were interested to get apps that would be able to perform multiple diseases diagnosis (multiplex). In both the focus group discussion and through questionnaire, respondents when asked about specific concerns, did not express serious concern about accuracy, sensitivity, data security, internet connectivity, smartphone capacity, apps design or update and ownership, indicating limited understanding of ongoing global changes in diagnostic field and potential limitation for their use in this area, but also indicating the need for training before deployment of these tools for public use and patient care.

Smartphone-enabled diagnostics show promise for enhancing healthcare in Tanzania. However, their adoption raises ethical concerns. It's crucial to obtain informed consent from

patients, addressing potential language, literacy, and cultural barriers. Patients should understand data usage, security, and privacy. Healthcare providers must ensure the accuracy of these tools, maintain transparency about limitations, and prioritize clinical validation. Equitable access across socioeconomic groups is vital to avoid widening healthcare disparities, addressing issues like digital literacy and affordability. Proper training for healthcare professionals is essential to integrate technology ethically, ensuring patient safety and upholding professional standards [38, 39]. The successful deployment of mobile phone-enabled diagnostics in Tanzania would require a comprehensive approach that addresses several key factors. Factors such as technological infrastructure development, healthcare systems evaluation, regulatory frameworks to ensure data privacy, quality control, and adherence to medical standards, protecting patient information and maintaining ethical practices [40].

## Future perspectives for mobile phone-enabled diagnostics in Tanzania

The future of mobile phone-enabled diagnostics in Tanzania holds great promise for transforming healthcare delivery by providing convenient, affordable, and efficient diagnostic solutions. By focusing on developing user-friendly applications, ensuring data security, fostering collaborations among key stakeholders, and investing in training programs, Tanzania can harness the full potential of mobile technology to advance healthcare services and improve health outcomes for its population [16, 41].

## Limitations of the study

A large proportion of respondents were from urban settings, which could have significantly influence knowledge, use, and ownership of smartphones. Urban populations often have greater access to technology and resources, potentially leading to higher levels of smartphone usage and familiarity with mobile health applications [42]. This urban bias may not accurately represent the situation in rural areas where access to accurate medical and veterinary diagnostics is more limited. Therefore, studies targeting rural populations are crucial, as they are often the most deprived of access to accurate medical and veterinary diagnostics of diseases. Given the disparities in access to healthcare resources between urban and rural areas, understanding the unique challenges and needs of rural communities is essential for ensuring equitable access to mobile phone-enabled diagnostics. Furthermore, this study solely targeted human and animal healthcare professionals, overlooking the perspectives of the general population. It is plausible that other population groups, such as patients and community members, may have different understandings and opinions on the use of smartphones for medical diagnostics. Their perspectives and experiences could significantly influence the acceptability and adoption of new technologies in healthcare settings. Future studies should aim to include a diverse range of participants to capture a comprehensive understanding of attitudes and perceptions towards smartphone-enabled diagnostics among different population groups. Moreover, this study did not collect information about the ethical, legal, and social implications of the use of smartphones for medical diagnosis, which are known barriers at different levels, including regulatory authorities. Ethical considerations, data privacy concerns, and regulatory frameworks play crucial roles in shaping the implementation and adoption of mobile health technologies [38, 43]. Therefore, future research endeavours should incorporate an assessment of these factors to inform policy development and ensure the responsible and ethical deployment of smartphone-based medical diagnostics in Tanzania.

## Conclusion

The advancement of mobile phone-enabled diagnostics offers transformative potential for healthcare accessibility in Tanzania. Smartphones, serving as analytical tools for point-of-care

testing, can deliver affordable, sensitive, and rapid medical solutions. Leveraging smartphone sensors and connectivity features like NFC, Bluetooth, and WiFi can bolster Tanzania's diagnostic capabilities, facilitating prompt disease management. With increasing mobile penetration and acceptance among healthcare professionals, the integration of smartphone-based diagnostic tools with advanced databases is poised for growth. By prioritizing user experience, functionality, and adherence to WHO's ASSURED criteria, Tanzania can foster universal access to cutting-edge molecular diagnostics. While our study indicates a favorable digital literacy among Tanzanian medical and veterinary professionals, targeted awareness campaigns and training are crucial to optimize the adoption and utilization of mobile diagnostic tools for enhanced healthcare outcomes.

## Supporting information

**S1 Questionnaire. The questionnaire used for collecting information for understanding the landscape of smartphone use in Tanzania and their potential application as m-health devices.** Our key objectives were: I. Understanding Digital Health for Infectious Disease in Low Resource Settings; II. Understanding Data Integrity and Security in Digital Health in Tanzania; III. Exploring the Relationship between Data and Healthcare Policy; IV. Mobile phone Devices for Data Acquisition and Communication in Tanzania; and V. Capacity Strengthening—Educational Training underpinning Mobile Health.
(DOCM)

## Acknowledgments

The authors would like to thank medical officers in-charge at all health facilities, which participated in the study for providing permission and time for their staff to take part in this study. We also extend our gratitude to our driver, Mr John Mushi, for careful driving across different regions and survey sites.

## Author Contributions

**Conceptualization:** Ndyetabura O. Theonest, John-Mary Vianney, Julien Reboud, Blandina T. Mmbaga, Jonathan M. Cooper, Joram Buza.

**Data curation:** Ndyetabura O. Theonest, Kennedy Ngowi, Allen Lyimo, Dina Machuve, John-Mary Vianney, Joram Buza.

**Formal analysis:** Ndyetabura O. Theonest, Kennedy Ngowi, Allen Lyimo, Dina Machuve.

**Funding acquisition:** Julien Reboud, Blandina T. Mmbaga, Jonathan M. Cooper, Joram Buza.

**Investigation:** Ndyetabura O. Theonest, Kennedy Ngowi, Elizabeth R. Kussaga, Allen Lyimo, Davis Kuchaka, Irene Kiwelu, Dina Machuve, John-Mary Vianney, Julien Reboud, Blandina T. Mmbaga, Jonathan M. Cooper, Joram Buza.

**Methodology:** Ndyetabura O. Theonest, Kennedy Ngowi, Elizabeth R. Kussaga, Allen Lyimo, Davis Kuchaka, Irene Kiwelu, Dina Machuve, John-Mary Vianney, Blandina T. Mmbaga, Joram Buza.

**Project administration:** Julien Reboud, Jonathan M. Cooper, Joram Buza.

**Resources:** Julien Reboud, Jonathan M. Cooper, Joram Buza.

**Supervision:** Blandina T. Mmbaga, Jonathan M. Cooper, Joram Buza.

**Validation:** Julien Reboud.

**Visualization:** Julien Reboud, Jonathan M. Cooper.

**Writing – original draft:** Ndyetabura O. Theonest, Joram Buza.

**Writing – review & editing:** Ndyetabura O. Theonest, Kennedy Ngowi, Elizabeth R. Kussaga, Allen Lyimo, Davis Kuchaka, Irene Kiwelu, Dina Machuve, John-Mary Vianney, Julien Reboud, Jonathan M. Cooper, Joram Buza.

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
