## [Decision Letter · Decision Letter 0]

3 Aug 2023

PDIG-D-23-00248

Status and future prospects for mobile phone-enabled diagnostics in Tanzania

PLOS Digital Health

Dear Dr. THEONEST,

Thank you for submitting your manuscript to PLOS Digital Health. After careful consideration, we feel that it has merit but does not fully meet PLOS Digital Health's publication criteria as it currently stands. Therefore, we invite you to submit a revised version of the manuscript that addresses the points raised during the review process.

Please submit your revised manuscript within 60 days Oct 02 2023 11:59PM. If you will need more time than this to complete your revisions, please reply to this message or contact the journal office at digitalhealth@plos.org. Please include the following items when submitting your revised manuscript:

We look forward to receiving your revised manuscript.

Kind regards,

Haleh Ayatollahi

Section Editor

PLOS Digital Health

Journal Requirements:

2. Please provide separate figure files in .tif or .eps format only and remove any figures embedded in your manuscript file. Please also ensure that all files are under our size limit of 10MB.

3. Some material included in your submission may be copyrighted. According to PLOS’s copyright policy, authors who use figures or other material (e.g., graphics, clipart, maps) from another author or copyright holder must demonstrate or obtain permission to publish this material under the Creative Commons Attribution 4.0 International (CC BY 4.0) License used by PLOS journals. Please closely review the details of PLOS’s copyright requirements here: PLOS Licenses and Copyright. If you need to request permissions from a copyright holder, you may use PLOS's Copyright Content Permission form.

Potential Copyright Issues:

Figure 1: please (a) provide a direct link to the base layer of the map (i.e., the country or region border shape) and ensure this is also included in the figure legend; and (b) provide a link to the terms of use / license information for the base layer image or shapefile. We cannot publish proprietary or copyrighted maps (e.g. Google Maps, Mapquest) and the terms of use for your map base layer must be compatible with our CC-BY 4.0 license. 

"

Additional Editor Comments (if provided):

The manuscript was interesting. Please address the following comments in your revision.

1- Please follow the journal instructions for referencing style.

2- Some terms such as smartphone are started with capital letters. It is better to revise the manuscript and be consistent in writing. 

3- The results section presents some simple descriptive statistics. I suggest the researchers to conduct more statistical analysis to find the correlation between different variables.

4- In Table 1, please use “sex” instead of “Gender”.

5- Please present the limitations of the study before the conclusion section.

Reviewers' comments:

Reviewer's Responses to Questions

**Comments to the Author**

1. Does this manuscript meet PLOS Digital Health’s publication criteria? Is the manuscript technically sound, and do the data support the conclusions? The manuscript must describe methodologically and ethically rigorous research with conclusions that are appropriately drawn based on the data presented.

Reviewer #1: Yes

Reviewer #2: Partly

2. Has the statistical analysis been performed appropriately and rigorously?

Reviewer #1: Yes

Reviewer #2: Yes

3. Have the authors made all data underlying the findings in their manuscript fully available (please refer to the Data Availability Statement at the start of the manuscript PDF file)?

Reviewer #1: No

Reviewer #2: No

4. Is the manuscript presented in an intelligible fashion and written in standard English?

Reviewer #1: Yes

Reviewer #2: No

5. Review Comments to the Author

Reviewer #1: Plos Digital Health manuscript review 

Status and future prospects for mobile phone-enabled diagnostics and Tanzania

This is an interesting study of smartphone use among veterinary and medical staff in Tanzania, with an emphasis on mHealth utilization.

The authors should consider paying for editing the paper for minor English grammar issues. Although the meaning is clear, there are some places where the grammar seems odd.

Abstract: Use headers in the abstract (Introduction, Methods, Results, Discussion, Conclusion). Please spell out UTI.

Data collection: Who were the interviewers? It's common to include the initials of the co-authors who did the interviews. Were the responses input into the software by the study subjects or by the interviewers?

Statistical analysis: Consider changing the first few words to read “Our analysis used a sequential explanatory design…” Who did the thematic analysis of the qualitative interviews? Were the recommended themes reviewed by other authors? How were final decisions made in cases of disagreement about the themes?

Table 1: Please spell out ICT.

Table 2: Consider putting the results under “Time of using smart phone per day” and “Duration of usage of mHealth applications” into numerical order. It’s a little confusing to have “over 5 hours” as the first response category and “5 hours” as the last response category. Similarly, it’s confusing to have “More than 5 Years” appear prior to “3 Years.”

Results: Please further describe the thematic analysis of the qualitative interviews and focus groups. Currently, there is only a listing of priority diseases, which appears to be the response from an open-ended question.

Discussion: Consider making the discussion section a bit less of a repetition of the results. Start with a short paragraph summarizing the main results. The comparison with other studies is interesting but does not require the results to be re-described in much detail. For Jones et al. 2011, please use a numbered citation. Please write out what L MICs means.

The “Limitation of the study” section should appear before the “Conclusion” section.

Data availability statement: This journal may require a specific statement as to availability, specifically, placement of data in a shared repository. Please check the Journal requirements.

Reviewer #2: It would have been beneficial to the reviewer to have had the structured questionnaires available along with the manuscript upon review. Please ensure that they are included in the final submission. Ensure that all relevant data is included and made available for public use upon publication of the manuscript. 

Included with this review is an attachment in MS Word doc format with the reviewer's recommendation and comments. As far as the manuscript being presented in an intelligible fashion and written in standard English, I have cited lines from the manuscript where grammatical corrections may be made to improve the flow of ideas written on paper. These are mere recommendations which the authors may use as guidelines for revision. 

Please read the reviewer's questions about certain areas of the paper, more importantly where quantitative, unstructured data may have been collected and the nature of how that information was coded, categorized, and summarized. Please include your procedures of how this was done if applicable.

6. PLOS authors have the option to publish the peer review history of their article (what does this mean?). If published, this will include your full peer review and any attached files.

**Do you want your identity to be public for this peer review?** For information about this choice, including consent withdrawal, please see our Privacy Policy.

Reviewer #1: No

Reviewer #2: No

---

## [Decision Letter · Decision Letter 1]

22 Jan 2024

PDIG-D-23-00248R1

Status and future prospects for mobile phone-enabled diagnostics in Tanzania

PLOS Digital Health

Dear Dr. THEONEST,

Thank you for submitting your manuscript to PLOS Digital Health. After careful consideration, we feel that it has merit but does not fully meet PLOS Digital Health's publication criteria as it currently stands. Therefore, we invite you to submit a revised version of the manuscript that addresses the points raised during the review process.

Please submit your revised manuscript within 60 days Mar 22 2024 11:59PM. If you will need more time than this to complete your revisions, please reply to this message or contact the journal office at digitalhealth@plos.org. Please include the following items when submitting your revised manuscript:

We look forward to receiving your revised manuscript.

Kind regards,

Haleh Ayatollahi

Section Editor

PLOS Digital Health

Journal Requirements:

Additional Editor Comments (if provided):

Reviewers' comments:

Reviewer's Responses to Questions

**Comments to the Author**

1. If the authors have adequately addressed your comments raised in a previous round of review and you feel that this manuscript is now acceptable for publication, you may indicate that here to bypass the “Comments to the Author” section, enter your conflict of interest statement in the “Confidential to Editor” section, and submit your "Accept" recommendation.

Reviewer #1: All comments have been addressed

Reviewer #3: (No Response)

Reviewer #4: All comments have been addressed

Reviewer #5: (No Response)

2. Does this manuscript meet PLOS Digital Health’s publication criteria? Is the manuscript technically sound, and do the data support the conclusions? The manuscript must describe methodologically and ethically rigorous research with conclusions that are appropriately drawn based on the data presented.

Reviewer #1: Yes

Reviewer #3: Partly

Reviewer #4: Yes

Reviewer #5: Partly

3. Has the statistical analysis been performed appropriately and rigorously?

Reviewer #1: Yes

Reviewer #3: I don't know

Reviewer #4: Yes

Reviewer #5: No

4. Have the authors made all data underlying the findings in their manuscript fully available (please refer to the Data Availability Statement at the start of the manuscript PDF file)?

Reviewer #1: Yes

Reviewer #3: Yes

Reviewer #4: Yes

Reviewer #5: Yes

5. Is the manuscript presented in an intelligible fashion and written in standard English?

Reviewer #1: Yes

Reviewer #3: Yes

Reviewer #4: Yes

Reviewer #5: No

6. Review Comments to the Author

Reviewer #1: No additional review comments. Excellent revision.

Reviewer #3: • The introduction outlines the need for information on smartphone use in disease diagnosis among healthcare professionals in Tanzania, but the objectives and hypotheses are not explicitly stated. Clearly defining these aspects.

• The study's sample size determination lacks detailed justification, and the distribution across regions appears arbitrary. The uneven distribution may affect the study's external validity and generalizability.

• The study predominantly focuses on urban areas, potentially leading to sampling bias. Including rural areas is crucial for a comprehensive understanding of smartphone utilization in different settings.

• The absence of a control group limits the ability to compare smartphone use among healthcare professionals with the general population, hindering a broader perspective.

• Add a few quotes or excerpts from the responses of healthcare professionals in the results and discussion sections.

• Make visual representations, such as word clouds, to present the thematic analysis findings in a more accessible format.

• The study briefly touches upon potential barriers such as awareness, policy guidelines, mobile data connectivity, and cultural practices influencing smartphone use for healthcare. However, a more in-depth exploration of these barriers, through qualitative interviews or additional survey questions, is needed.

• While the results are presented, there is a lack of critical discussion regarding the implications of the findings. For instance, the low utilization of smartphones for healthcare practices raises questions about the underlying reasons, and potential strategies to address these issues should be discussed.

• In Figure 1, please include a link to the terms of use or license information. This information is crucial for meeting the CC-BY 4.0 license requirements.

• The methods section mentions thematic analysis, but details on the rigor, inter-rater reliability, or validation processes are lacking. A more robust description of the analytical approach would enhance the study's credibility. Also, you mentioned thematic analysis of open-ended responses, but the specific themes and subthemes identified from the responses are not explicitly presented. Providing a brief summary of the thematic analysis results could enhance clarity.

• The study briefly mentions ethical approval and written consent but lacks discussion on potential ethical considerations associated with smartphone-enabled diagnostics, such as data security, patient privacy, and informed consent for diagnostic applications.

• The study presents priority diseases for both human and animal healthcare professionals separately. A comparative analysis to identify commonalities and differences in priorities could provide valuable insights for developing integrated diagnostic solutions.

• Given the interdisciplinary nature of healthcare, emphasizing the potential benefits of multi-disciplinary collaboration in the discussion could highlight the holistic approach needed for successful implementation of mobile phone-enabled diagnostics.

• While limitations are briefly mentioned, a more detailed discussion of the study's limitations is needed. For example, discussing the potential biases introduced by the predominantly urban sample and the exclusion of the general population could be valuable.

• Concluding the paper with a section on potential future research directions could provide insights for researchers and policymakers. For instance, discussing the need for studies focusing on rural populations, exploring ethical considerations, and addressing regulatory challenges would contribute to the discussion.

• A dedicated conclusion section summarizing the key findings, implications, and potential next steps will provide a more conclusive ending to the paper.

Reviewer #4: All comments have been addressed and I am satisfied.

Reviewer #5: The concept of the study seems interesting as laying groundwork for next stages of digital engagement for Tanzania. There are still several typographical errors and grammatical inconsistencies, but the revision has addressed many of them. Conceptually there also seems to be some lack of cohesion with the introduction through to the conclusions which could use refinement. 

The author summary should not be a copy and pasted abstract starting in the second sentence and ending with the second to last, but rather how this work fits into the broader context of similar literature. The words "considerable disparity" carries connotations related to unfair treatment, would reconsider its use.

Introduction: would benefit from some substantiation of the value of mobile diagnostics. There is a sentence describing some examples, but with very few details. More information about the effectiveness of these tools in other areas or uptake of use would be helpful to understand the study's context.

Methods: There were three modalities of data collection (questionnaires, in-depth interview, and focus groups), but how did these integrate with each other? What was sent to the Clinical Research Institute, a transcribed interview with the questionnaires or a post-processed form of theme assignment by the investigators? Table 2 seems like it is reporting the questionnaire entirely, what additional information did the interviews provide?

Methods, study area: what does "professionals based on literacy level providing constructive opinion on the use of mobile phone for health purposes" indicate? Did the veterinary clinics not have this? 

Methods, study subjects: Regarding the statement "based on their professional skills and their role at the health facilities," (lines 160-161) does this imply the selection wasn't based on professional licensing status?

Methods: The power calculation was not clear to me. 

Methods, data collection: the author initials and names listed in the manuscript may be more appropriate in the supplemental materials, but this may be a style difference. 

Methods: there should be explicit statement of the primary and (if applicable) secondary outcomes. These seem to be mentioned tangentially in the data collection section. 

Methods, statistical analysis: I would change the language of the analysis to be directly related to your outcomes. "To investigate the changes and trends on awareness and literacy and use of smartphone" does not provide enough information--is this referring to the logistic regression and awareness of mHealth specifically? 

There is not a mention of mHealth prior to writing about the logistic regression. mHealth is bigger than simple medical diagnostic tools, so this should be differentiated if awareness of this tool is to be a dedicated outcome. 

The statement about results in the statistical analysis section could be removed. The statistical analysis section has some sentences that seem out of order for the logical sequence.

Results: 

The sentence "workers responded to a survey which aimed to assess their perception towards the use of smartphone in clinical practice for diagnosis of diseases/conditions" belongs in the methods section. 

Table 1 formatting should be consistent. 

Table 2 formatting should be consistent: font and indentations are variable. 

There is mention of the various uses of smartphones in healthcare practice, including accessing guidelines and patient education materials. If it was studied and is being reported, the distribution should be quantified. It may be helpful to include in Table 2, if so. 

Changing the term "priority diseases" to "priority diseases/pathophysiology" still implies that pregnancy itself is a problem or harmful. If the domain is not referring to "complications of pregnancy," then the label should be changed. 

In lines 267-272, this is an important foundation for why the specific phone apps were reported at all, but it needs more development for me to agree with the relevance. 

Some grammatical and typographical points to address (not comprehensive): 

- The word "diagnostic" is an adjective and "diagnostics" is a noun. The phrase should either be "Point of care diagnostic technologies" or "Point of care diagnostics." 

- "focussed group" should be "focus group" 

- In lines 34-35 of the abstract it is not immediately clear that the numbers following the region names are the distribution of participants

- some semicolons are not used correctly

- line 38, the number 97.4% should be in parentheses 

- The abstract discussion is a very long run-on sentence

- The abstract conclusion is also a very long run-on sentence.

- lines 83-84 the word "diseases" should be singular, or as applicable if the word "pathophysiology" was meant to be included

- line 96, "easy" should be "ease"

- line 116, "indepth" should be hyphenated

- "m-health" is typically abbreviated "mHealth," but is also not consistent throughout the manuscript 

- line 133, the word "brackets" should be "parentheses" 

- line 151 the word "proportional" should be "proportion" 

- line 212, the word "animals" should be singular

7. PLOS authors have the option to publish the peer review history of their article (what does this mean?). If published, this will include your full peer review and any attached files.

**Do you want your identity to be public for this peer review?** For information about this choice, including consent withdrawal, please see our Privacy Policy. 

Reviewer #1: No

Reviewer #3: No

Reviewer #4: Yes: Abdulsalam S. Mustafa

Reviewer #5: Yes: Catherine Bielick

---

## [Decision Letter · Decision Letter 2]

24 May 2024

PDIG-D-23-00248R2

Status and future prospects for mobile phone-enabled diagnostics in Tanzania

PLOS Digital Health

Dear Dr. THEONEST,

Thank you for submitting your manuscript to PLOS Digital Health. After careful consideration, we feel that it has merit but does not fully meet PLOS Digital Health's publication criteria as it currently stands. Therefore, we invite you to submit a revised version of the manuscript that addresses the points raised during the review process.

Please submit your revised manuscript within 30 days Jun 23 2024 11:59PM. If you will need more time than this to complete your revisions, please reply to this message or contact the journal office at digitalhealth@plos.org. Please include the following items when submitting your revised manuscript:

We look forward to receiving your revised manuscript.

Kind regards,

Haleh Ayatollahi

Section Editor

PLOS Digital Health

Journal Requirements:

Additional Editor Comments (if provided):

Reviewers' comments:

Reviewer's Responses to Questions

**Comments to the Author**

1. If the authors have adequately addressed your comments raised in a previous round of review and you feel that this manuscript is now acceptable for publication, you may indicate that here to bypass the “Comments to the Author” section, enter your conflict of interest statement in the “Confidential to Editor” section, and submit your "Accept" recommendation.

Reviewer #1: All comments have been addressed

Reviewer #4: All comments have been addressed

Reviewer #5: (No Response)

2. Does this manuscript meet PLOS Digital Health’s publication criteria? Is the manuscript technically sound, and do the data support the conclusions? The manuscript must describe methodologically and ethically rigorous research with conclusions that are appropriately drawn based on the data presented.

Reviewer #1: Yes

Reviewer #4: Yes

Reviewer #5: Yes

3. Has the statistical analysis been performed appropriately and rigorously?

Reviewer #1: Yes

Reviewer #4: Yes

Reviewer #5: Yes

4. Have the authors made all data underlying the findings in their manuscript fully available (please refer to the Data Availability Statement at the start of the manuscript PDF file)?

Reviewer #1: Yes

Reviewer #4: Yes

Reviewer #5: Yes

5. Is the manuscript presented in an intelligible fashion and written in standard English?

Reviewer #1: Yes

Reviewer #4: Yes

Reviewer #5: Yes

6. Review Comments to the Author

Reviewer #1: No further comments. The authors have adequately addressed reviewer comments.

Reviewer #4: I am satisfied with the comments and recommend the manuscript for acceptance in this journal.

Reviewer #5: The substance of most comments has been addressed. Thank you for your attention to these details. There are still some areas requiring attention and typographical errors remaining. 

Introduction: 

107 - POC should have a following noun like "POC test" or "POC enzyme linked immunosorbent assay" without the "for"

140 - correct the double space 

Methods: 

- Thank you for stating the variable of interest more directly in lines 232-233, but the sentence is still not clear due to the incorrect semicolon use. It should read something more like "The primary outcome was whether or not a person was familiar with mobile phone-enabled diagnostics." Why was both a binary and multinomial logistic regression model created? Either the primary outcome was binary or it was multi-class, but it can't be both. If two types of models were created for two different outcomes, then the second outcome also needs to be stated directly. 

- 232-235 should be broken up into more than one sentence 

- The term mHealth should be defined before line 154 if it's going to be used interchangeably with POC diagnostics, probably in the Introduction.

- 201 Letters should not be capitalized

- 213 "to ascertain on awareness" is not the correct usage and has two spaces to correct 

Results: 

Table 1 still has inconsistent formatting. The category labels do not need to be in the middle of the table. 

Table 3 title: would state the outcome in the title, the words "correlation" and "different parameters" are redundant, "mhealth" is inconsistent with prior usage

Table 3 is not referenced in the text, the results are also not discussed

Discussion

The implications of the findings for the logistic regression model(s) should be included. 

342, 357, 402 the "-" is not needed

361 "mobile health (mHealth)" should be defined earlier 

362 "LMICs" has already been defined as an abbreviation

363 inconsistent formatting with the word "mhealth"

367-368 "mobile health (mHealth)" has already been defined 

Tables and Figures don't need to be referenced with parentheses in the discussion section, just the results section 

415 "condition" should be plural 

451 "Limitation" should be plural, "study" should be capitalized

7. PLOS authors have the option to publish the peer review history of their article (what does this mean?). If published, this will include your full peer review and any attached files.

**Do you want your identity to be public for this peer review?** For information about this choice, including consent withdrawal, please see our Privacy Policy. 

Reviewer #1: No

Reviewer #4: Yes: Abdulsalam Salihu Mustafa

Reviewer #5: Yes: Catherine Bielick

---

## [Editor Report · Decision Letter 3]

28 Jun 2024

Status and future prospects for mobile phone-enabled diagnostics in Tanzania

PDIG-D-23-00248R3

Dear Dr THEONEST,

We are pleased to inform you that your manuscript 'Status and future prospects for mobile phone-enabled diagnostics in Tanzania' has been provisionally accepted for publication in PLOS Digital Health.

Best regards,

Haleh Ayatollahi

Section Editor

PLOS Digital Health